

# Identification of potential genes in upper tract urothelial carcinoma using next-generation sequencing with bioinformatics and in vitro analyses

Hsiang-Ying Lee[1,2,3,4], Ching-Chia Li[3,4,5], Wei-Ming Li[3,4,5,6], Ya-Ling Hsu[5], Hsin-Chih Yeh[2,3,4,5], Hung-Lung Ke[3,4,5], Bi Wen Yeh[3,4], Chun-Nung Huang[2,3,4,5], Chien-Feng Li[7], Po-Lin Kuo[1,8,9] and Wen-Jeng Wu[3,4,5,9]

[1] Graduate Institute of Clinical Medicine, College of Medicine, Kaohsiung Medical University, Kaohsiung, Taiwan

[2] Department of Urology, Kaohsiung Municipal Ta-Tung Hospital, Kaohsiung, Taiwan

[3] Department of Urology, Kaohsiung Medical University Hospital, Kaohsiung, Taiwan

[4] Department of Urology, School of Medicine, College of Medicine, Kaohsiung Medical University, Kaohsiung, Taiwan

[5] Graduate Institute of Medicine, College of Medicine, Kaohsiung Medical University, Kaohsiung, Taiwan

[6] Department of Urology, Ministry of Health and Welfare Pingtung Hospital, Pingtung, Taiwan

[7] Department of Pathology, Chi Mei Medical Center, Tainan, Taiwan

[8] Center for Biomarkers and Biotech Drugs, Kaohsiung Medical University, Kaohsiung, Taiwan

[9] Institute of Medical Science and Technology, National Sun Yat-Sen University, Kaohsiung, Taiwan

Corresponding author
Wen-Jeng Wu,
ashum1009@gmail.com

## ABSTRACT

**Background**. We aimed to identify prognostic biomarkers of upper tract urothelial carcinomas (UTUCs), including microRNAs (miRNAs) and genes which account for only 5% to 10% of all urothelial carcinomas (UCs). In Taiwan, this figure is markedly higher, where it can reach up to 30% of UC cases.

**Materials and Methods**. Using next-generation sequencing (NGS), we analyzed two pairs of renal pelvis tumors and adjacent normal urothelial tissues to screen miRNAs and messenger RNAs. By combining bioinformatics analysis from miRmap, Gene Expression Omnibus (GEO), and Oncomine and Ingenuity® Pathway Analysis databases, we identified candidate genes. To search for upstream miRNAs with exact target binding sites, we used miRmap, TargetScan, and miRDB to enforce evidence. Then, we clarified gene and protein expression through an in vitro study using western blot analysis and quantitative real-time reverse transcriptase-PCR.

**Results**. Interactions between selected target genes obtained using the NGS and miRmap methods were assessed through a Venn diagram analysis. Six potential genes, namely, PDE5A, RECK, ZEB2, NCALD, PLCXD3 and CYBRD1 showed significant differences. Further analysis of gene expression from the GEO dataset indicated lower expression of PDE5A, RECK, ZEB2, and CYBRD1 in bladder cancer tissue than in normal bladder mucosa, which indicated that PDE5A, RECK, ZEB2, and CYBRD1 may act as tumor suppressors in UTUC. In addition, we compared the expression of these genes in various UC cell lines (RT4, BFTC905, J82, T24, UMUC3, 5637, BFTC 909, UMUC14) and found decreased expression of PDE5A in muscle-invasive UC cells compared with the RT4 cell line. Furthermore, by using paired UTUC and normal

tissues from 20 patients, lower PDE5A expression was also demonstrated in tumor specimens.

**Conclusions**. Our findings suggest these candidate genes may play some roles in UTUC progression. We propose that these markers may be potential targets clarified by in vitro and in vivo experiments. PDE5A also potentially presents tumor suppressor genes, as identified by comparing the expression between normal and tumor specimens.

# INTRODUCTION

Urothelial carcinomas (UCs) are derived from the urothelium and divided into upper and lower tract UCs according to tumor location. Upper tract urothelial carcinoma (UTUC), originating from the renal pelvis and ureter presents unique features in Taiwan compared to those in the Western regions. UTUC accounts for only 5%–10% of all UCs in Western countries but up to approximately 30% of UCs in Taiwan (*Li et al., 2016*). In addition, according to a recent report from the Taiwan Cancer Registry in 2013, the age-standardized incidence per 100,000 person-years was 4.09% and 4.37% in men and women, respectively, with a male–to-female ratio of 1:1.2 (*Shen et al., 2017*). Conversely, *Shariat et al. (2011)* calculated that the incidence in men is twice that in women. Specific environmental risk factors in Taiwan have been cited, including black foot disease and Chinese herb nephropathy (*Colin et al., 2009*; *Miyazaki & Nishiyama, 2017*). However, although herbs with aristolochic acid components have been prohibited and environmental contamination by arsenic has improved, the incidence of UTUC has not decreased.

Radical nephroureterectomy with bladder cuff excision is the gold standard treatment for UTUC. Despite advances in diagnostic imaging and surgical techniques, the oncologic outcomes of five-year recurrence-free survival and cancer-specific survival vary greatly (*Margulis et al., 2009*; *Peyronnet et al., 2017*). Some pathological variables including tumor stage, grade, lymph node metastasis status, lymphovascular invasion, tumor architecture, and multifocality are considered influential factors in determining survival (*Lee et al., 2015*). However, these pathological characteristics still possess only a limited ability to predict outcomes. Moreover, the prognosis of patients with locally advanced or metastatic UTUC is poor, and chemotherapy lacks substantial benefits for high-risk cancer (*Patel et al., 2014*). Therefore, specific biomarkers must be identified to predict outcomes and tailor personalized treatment and surveillance strategies.

MicroRNAs (miRNAs) are a family of short (22nt long, on average) small noncoding RNAs. In UTUC, miRNAs that play a major role in tumor progression and cancer-specific survival were identified as regulators of gene expression, including oncogene and tumor suppressor genes. Signalling pathways modified by prognostic markers have been recognized as being involved in UTUC progression (*Izquierdo et al., 2017*). The

regulatory roles of miRNAs in cell growth, apoptosis, autophagy, and aging have also been implicated (*Sohn, 2018*).

Next-generation sequencing (NGS) is a useful tool for evaluating the genomic characteristics of various cancers, such as lung adenocarcinoma, renal cell carcinoma, colon cancer, and gastric cancer (*Hsu et al., 2017*; *Youssef et al., 2017*; *Chen et al., 2017*). Through high-throughput sequencing, considerably high or low expression of messenger RNAs and small RNAs can be identified. The purpose of this study was to discover valuable biomarkers using a combination of bioinformatics tools including miRmap (*Vejnar, Blum & Zdobnov, 2013*), Gene Expression Omnibus (GEO), Oncomine (*Rhodes et al., 2007*), Ingenuity® Pathway Analysis (IPA) (*Zhao, Liu & Qu, 2017*), TargetScan, and miRDB (*Wong & Wang, 2015*) in our UTUC patient specimens focused on renal pelvis cancer. Prior to the study findings were expected to reveal potential biomarkers and molecular mechanisms to determine suitable diagnostic and therapeutic strategies.

## MATERIAL AND METHODS

### Clinical UTUC specimen characteristics

We collected two pairs of tumor specimens and normal urothelial tissues from two UTUC patients in our hospital. These two patients had similar background characteristics, they were both female with a diagnosis of high-grade renal pelvis UC (Table 1). The Kaohsiung Medical University Hospital granted ethical approval to conduct the study within its facilities. This study was approved by our institutional review board (KMUHIRB-E(I)-20170018, KMUH107-7R56), and all patients provided written consent. To ensure the quality of the specimens, we collected them following strict procedures. After performing radical surgery, we collected specimens within 30 min then immediately preserved them in a −195.79 °C liquid nitrogen container.

### Next-generation sequencing

Using high-throughput sequencing and, NGS, we analyzed the whole genome, including mRNAs and microRNAs (*Krämer et al., 2014*), and the sequencing depth is 30 million reads per sample. Two pairs of renal pelvis tumors and adjacent normal specimens were used in this study. Total RNA was extracted using TRIzol® Reagent (Invitrogen, Waltham, MA, USA) according to the manufacturer's instructions. Using an ND-1000 spectrophotometer (NanoDrop Technologies, Wilmington, DE, USA), the purified RNA was quantified at OD260nm and qualitatively analyzed using a Bioanalyzer 2100 (Agilent Technologies, Santa Clara, CA, USA) with an RNA 6000 LabChip kit (Agilent Technologies, Santa Clara, CA, USA). We cooperated with the Welgene Biotechnology Company (Welgene, Taipei, Taiwan) to assess RNA-seq and small RNA-seq using the official protocol of Illumina (San Diego, CA, USA). Raw sequences were obtained using the Illumina Pipeline software bcl2fastq v2.0.

After obtaining qualified reads, trimmomatics software was used to trim or remove the reads according to the quality score using TopHat/Cufflinks. A fold change >2 and fragments per kilobase million (FPKM) >0.3 were defined as significant expressions for mRNA analysis and a fold change >2 and reads per million (RPM) >1 were defined as

**Table 1** The characteristics of two patients whose specimens were for NGS analysis.

|  | Patient 1 | Patient 2 |
| --- | --- | --- |
| Gender | Female | Female |
| Age | 67 | 76 |
| Tumor site | Renal pelvis | Renal pelvis |
| Laterality | Left | Left |
| Tumor grading | high | high |
| Pathology T stage | T3 | T1 |
| N stage | N0 | N0 |
| M stage | M0 | M0 |
| Lymphovascular invasion | No | No |
| Perineural invasion | No | No |

significant expressions for miRNA analysis. The $p$-value was calculated by cuffdiff with non-grouped sample using "blind mode", in which all samples were treated as replicates of a single global "condition" and used to build one model for statistical testing. The Benjamini and Hochberg method was used to obtain $q$-values after adjusting for the false discovery rate. Therefore, we utilized these methods to diminish the bias of workflow performance if the sample size was small as in previous research (*Baccarella et al., 2018*). We have deposited our data in the Gene Expression Omnibus (GEO) repository with the accession number GSE159824.

## miRmap database analysis for microRNA target predicting

miRmap is a web application offering resource-predicting corresponding targets, including miRNAs or genes. The miRmap library offers a complete range of features by combining thermodynamic, evolutionary, probabilistic, and sequence-based features to estimate potential candidate targets. By calculating miRNA–mRNA interaction prediction scores, and ranking them as miRmap scores, the most possible biomarker can be identified. In our study, we used final putative targets with miRmap scores $\geq$ 99.0.

## Gene Expression Omnibus database and statistical analysis

The GEO database is a web platform comprising microarrays, NGS, and other high-throughput functional genomics submitted by the research community. In the present study, we chose GSE19915 expression profiling using arrays, published in 2010. This database includes 144 bladder cancer samples and 12 normal samples analyzed using microarrays with the Strata gene Universal Human Reference RNA as the common reference sample. We then extracted raw data from GEO2R (https://www.ncbi.nlm.nih. gov/geo/geo2r/?acc=GSE19915) and calculated the expression differences using SPSS, version 19 (IBM Corp., Somers, NY, USA). The $p$ value was calculated using Student's $t$-test, and $p < 0.05$ was considered statistically significant.

## Oncomine database analysis

The Oncomine bioinformatics database collects over 18,000 cancer gene expression microarrays spanning various types of cancer including bladder cancer. Comparisons

between clinical bladder cancer and normal specimens were performed using raw data on mRNA expression. A *p* value <1E-4, fold change >2, and gene rank in the top 10% were defined as inclusion criteria. We then calculated the *p* value using the Oncomine database with a two-sided Student's *t*-test.

## Ingenuity®Pathway Analysis

IPA is a web-based software that analyzes omics data. An interaction network was presented by integrating miRNAs and candidate genes. From the gene expression dataset, upstream factors and probable downstream effects were elucidated.

## TargetScan and miRDB analyses

Using the online resources TargetScan and miRDB, we identified functional miRNAs from candidate gene targets. These computational tools can recognize the compensatory binding sites of matched targets from each miRNA.

## Western blot analysis

The cell lysates (30 μg) from urothelial cancer cell lines (RT4, BFTC905, J82, T24, UMUC3, BFTC 909 were obtained from Bioresource Collection and Research Center (BCRC), 5637 from ATCC, UMUC14 from European Collection of Authenticated Cell Cultures (ECACC)) were separated by 8%–12% SDS–PAGE gel and transferred onto a polyvinylidene difluoride membrane. Antibodies against human ZEB2 (1:5000 dilution, 14026-1-AP, Proteintech), CYBRD1 (1:5000 dilution, 26735-1-AP, Proteintech), PDE5A (1:5000 dilution, 22624-1-AP, Proteintech), and GAPDH (1:20000 dilution, 60004-1-Ig, Proteintech) were employed as primary antibodies. Rabbit or mouse IgG antibodies coupled with horseradish peroxidase were used as secondary antibodies. An enhanced chemiluminescence kit (Amersham) was used for protein detection (Amershan Biosciences, Piscataway, NJ, USA)

## Quantitative real time reverse transcriptase-PCR (qRT-PCR)

The mRNA expressions levels of PDE5A, CYBRD1, and ZEB2 were analyzed using qRT-PCR. Total RNA was isolated from RT4, BFTC905, J82, T24, UMUC3, 5637, BFTC 909, UMUC14, and human tissues using an RNAzol kit (TEL-TEST Inc., Friendswoods, TX, USA). For reverse transcription, 5 μg of total RNA was used with a Thermoscript reverse transcriptase (RT)-PCR system. Gene expression levels were amplified and detected using a Power SYBR®Green PCR Master Mix kit (Applied Biosystems; Life Technologies, Inc.) in a QuantStudio$^{TM}$ 3 Real-Time PCR Detection System (Applied Biosystems; Life Technologies, Inc.).The PCR products were size-fractionated by electrophoresis in agarose gel, stained with TOOLS DNA View (TOOLS, Taipei, Taiwan) and photographed by ultraviolet light illumination.Primers used for target gene expression in the qRT-PCR are as follows: PDE5A (sense primer: 5′-CGGCCCAAACCCTTAAAATT-3′; antisense primer: 5′-AGCGCTGTTTCCAGATCAGA-3′); CYBRD1(sense primer: 5′-CATGGTCACCGGCTTCGT-3′; antisense primer: 5′-CAGGTCCACGGCAGTCTGTA-3′); ZEB2(sense primer: 5′-ATATGGTGACACACAAGCCAGGGA-3′; antisense primer: 5′-GTTTCTTGCAGTTTGGGCACTCGT-3′).
## Immunohistochemistry

Immunohistochemical staining of PDE5A, CYBRD1, and ZEB2 was performed on 4-μm thick paraffin-embedded tissue sections. After deparaffinization, rehydration, antigen retrieval, and nonspecific binding blocking, tissue sections were incubated with an antibody against PDE5A (1:200 dilution, 22624-1-AP, Proteintech), CYBRD1 (1:200 dilution, 26735-1-AP, Proteintech), and ZEB2 (1:200 dilution, 14026-1-AP, Proteintech) at 37 °C for 1 h. The tissue sections were then rinsed twice with PBS for 2 min and then incubated with Mouse/Rabbit Probe HRP Labeling (BioTnA, TAHC03D) for 30 min at room temperature. They were then rinsed twice with PBS for 2 min and incubated in DAB peroxidase substrate solution (BioTnA) for 3 min, followed by a brief rinse with distilled water. The sections were counterstained with hematoxylin solution for 30 s (BioTnA, TA01NB). The results were recorded using a magnifier digital camera. Controls included omitting or preabsorbing the primary antibody and omitting secondary antibody.

## RESULTS

### Identification of significant miRNA expression in renal pelvis cancer

To study potentially influential miRNA–mRNA interactions, miRNAs with a fold change >2 and FPKM >0.3 were selected. In total, 54 upregulated and 9 downregulated miRNAs were identified (Tables 2 and 3). The heatmap shows significant differentially expressed miRNAs with z-score (log2) values between tumor and normal samples by using color clustering on the Morpheus web-tool. Blue represents downregulation and red represents upregulation (Fig. 1).

### Identification of significant miRNA–mRNA interactions and genes

To identify important miRNA–mRNA interactions and genes, we simultaneously analyzed our NGS results and miRmap putative genes. Using the NGS platform, 86 putative upregulated mRNAs and 225 putative downregulated mRNAs were identified. Using miRmap techniques to screen genes with miRmap scores >99.0, which indicated high predictive strength of repression, 115 putative upregulated target genes and 470 putative downregulated target genes were identified. The overlapping genes between miRNA putative targets and differentially expressed genes of our dataset were identified by Venn diagram analysis, and six potential genes, namely, PDE5A, RECK, ZEB2, NCALD, PLCXD3, and CYBRD1 had significant differences (Fig. 2). To confirm whether the expression of these six identified genes was of clinical significance, we searched the GEO database for urothelial carcinoma or UTUC specimens. GSE19915 was chosen for the array. This array compared urothelial carcinoma specimens and normal urothelial tissues (*Lindgren et al., 2010*). When these six genes were searched for in this database, PDE5A, RECK, ZEB2, and CYBRD1 showed significant differences between tumor tissue and normal tissue (all $p < 0.001$) (Fig. 3). The results demonstrated low expression of PDE5A, RECK, ZEB2, and CYBRD1 in urothelial carcinoma tissue. The four downregulated genes with cooresponding miRNAs are listed in Table 4. We further investigated these genes using the Oncomine database, and the results also indicated lower expression of PDE5A, RECK, ZEB2, and CYBRD1 in bladder cancer tissue than in normal bladder mucosa (Fig. 4).

## Identification of differentially expressed genes in UTUC and the potentially involved pathways

To understand which pathways were potentially involved in the expression of the four identified genes, we used IPA software for analysis. The miRNA-gene network cell cycle analysis revealed that all the aforementioned matched putative genes were associated with TP53. Four networks including urothelial carcinoma, invasion, metastasis, and urogenital cancer, were used to identify the affected genes. PDE5A, RECK, ZEB2, and CYBRD1 play essential roles by interacting with other miRNAs and genes in urothelial carcinoma networks which are represented by purple circles. RECK is an important factor in cancer invasion. RECK and ZEB2 are involved in cancer metastasis and are also associated with urogenital cancer networks (Figs. 5A–5D). Various miRNAs affect downstream mRNAs by interrupting cell cycle pathways and promoting cancer progression. The network of PDE5A-related molecules obtained using the Overlay tool in IPA included miR-19b-3p and WDR24.

## Identifying correlations among PDE5A, CYBRD1, and ZEB2 by an in vitro study

We compared the expression of these genes in various UC cell lines (RT4, BFTC905, J82, T24, UMUC3, 5637, BFTC 909, UMUC14) by western blot analysis and found decreased expression of PDE5A in muscle-invasive UC cells compared with the RT4 cell line (Fig. 6). As shown in Fig. 7, J82, T24, 5637, and BFTC 909 cells exhibited higher mRNA expression of CYBRD1 and ZEB2 but lower expression of PDE5A than RT4 cells. Protein expression was validated through histopathological analysis of our UTUC specimens, which revealed different levels and various localizations of PDE5A, CYBRD1, and ZEB2 present at different staining sites (Fig. 8) in 200X magnification. To further clarify whether PDE5A is associated with UC, we evaluated PDE5A expression through real-time PCR from the paired UTUC tissues (normal–tumor) of 20 patients, in total, 40 samples. The findings demonstrated lower expression of PDE5A in tumour specimens than in normal tissues. (Fig. 9) The clinical characteristics of the 20 patients are shown in Table 5.

## DISCUSSION

UTUC originates from the urothelium and spans the whole urinary tract. Although bladder cancer is also derived from the urothelium, aggressive UTUC possesses many characteristics different from those of bladder UC. Overall, 60% of UTUCs present invasive patterns at diagnosis compared with 15%–25% of bladder tumors (Rouprêt et al., 2018). However, the factors that contribute to the differences in UTUC rates between Western regions and Taiwan remain unclear. It is crucial to identify novel biomarkers that can compensate for or replace current surveillance techniques. We initially combined high-throughput NGS analysis with bioinformatics tools to select critical miRNAs or genes including PDE5A, RECK, CYBRD1, and ZEB2 for further comprehensive research.

Phosphodiesterase 5A (PDE5A), a gene responsible for coding cGMP-binding and cGMP-specific phosphodiesterase, is a member of the cyclic nucleotide phosphodiesterase family. The function of PDE5A is to degrade cGMP to 5′-GMP. Some studies have

**Table 2   Up expression of miRNA in renal pelvis cancer compared to adjacent normal tissue using next-generation sequencing.**

| miRNA name | precursor | Fold Change T1/N1 | Fold Change T2/N2 | Direction of change |
|---|---|---|---|---|
| hsa-miR-128-3p | hsa-mir-128-2 | 5.393939394 | 2.482288828 | Up |
|  | hsa-mir-128-1 | 5.74426413 | 2.427990236 | Up |
| hsa-miR-200a-5p | hsa-mir-200a | 7.921787709 | 5.284688995 | Up |
| hsa-miR-200a-3p | hsa-mir-200a | 10.04772448 | 4.169089518 | Up |
| hsa-miR-200b-3p | hsa-mir-200b | 9.661833489 | 5.984396618 | Up |
| hsa-miR-200c-3p | hsa-mir-200c | 12.07910337 | 6.287042777 | Up |
| hsa-miR-181c-3p | hsa-mir-181c | 3.812911726 | 6.021377672 | Up |
| hsa-miR-452-5p | hsa-mir-452 | 5.802035153 | 5.31097561 | Up |
| hsa-miR-210-3p | hsa-mir-210 | 24.96411483 | 6.72866242 | Up |
| hsa-miR-1307-3p | hsa-mir-1307 | 4.069922309 | 6.355513308 | Up |
| hsa-miR-130b-3p | hsa-mir-130b | 9.120198265 | 6.013647643 | Up |
| hsa-miR-149-5p | hsa-mir-149 | 7.352219075 | 5.438751472 | Up |
| hsa-miR-106b-5p | hsa-mir-106b | 6.602572783 | 2.809605489 | Up |
| hsa-miR-421 | hsa-mir-421 | 5.075050033 | 2.636952998 | Up |
| hsa-miR-345-5p | hsa-mir-345 | 4.386983632 | 6.092846271 | Up |
| hsa-miR-454-3p | hsa-mir-454 | 3.7627829 | 2.296692607 | Up |
| hsa-miR-1307-5p | hsa-mir-1307 | 4.343764381 | 3.805475504 | Up |
| hsa-miR-1260b | hsa-mir-1260b | 3.885969522 | 3.145619087 | Up |
| hsa-miR-1260a | hsa-mir-1260a | 3.365115228 | 2.901953557 | Up |
| hsa-miR-181d-5p | hsa-mir-181d | 4.586446105 | 6.408422725 | Up |
| hsa-miR-20a-5p | hsa-mir-20a | 11.61530945 | 6.656237624 | Up |
| hsa-miR-17-5p | hsa-mir-17 | 10.48332695 | 4.792083013 | Up |
| hsa-miR-425-5p | hsa-mir-425 | 8.732341228 | 7.380424454 | Up |
| hsa-miR-34a-5p | hsa-mir-34a | 3.420600858 | 2.170505872 | Up |
| hsa-miR-429 | hsa-mir-429 | 7.733315748 | 4.339373602 | Up |
| hsa-miR-301a-3p | hsa-mir-301a | 3.221881391 | 4.98474609 | Up |
| hsa-miR-941 | hsa-mir-941-1 | 3.413867323 | 2.183439332 | Up |
|  | hsa-mir-941-2 | 3.413867323 | 2.183439332 | Up |
|  | hsa-mir-941-3 | 3.413867323 | 2.183439332 | Up |
|  | hsa-mir-941-4 | 3.413867323 | 2.183439332 | Up |
|  | hsa-mir-941-5 | 3.413867323 | 2.183439332 | Up |
| hsa-miR-769-5p | hsa-mir-769 | 2.031387504 | 2.621387596 | Up |
| hsa-miR-335-3p | hsa-mir-335 | 6.192922017 | 5.23965448 | Up |
| hsa-miR-222-3p | hsa-mir-222 | 2.235965788 | 5.080788558 | Up |
| hsa-miR-203a-3p | hsa-mir-203a | 21.84072277 | 5.781700835 | Up |
| hsa-let-7d-5p | hsa-let-7d | 3.475411219 | 3.417895946 | Up |
| hsa-miR-181c-5p | hsa-mir-181c | 3.920424636 | 5.225985327 | Up |
| hsa-miR-93-5p | hsa-mir-93 | 5.664360678 | 3.228353016 | Up |
| hsa-miR-221-3p | hsa-mir-221 | 2.414195555 | 3.433672238 | Up |
| hsa-miR-151a-5p | hsa-mir-151a | 4.45419659 | 2.681035029 | Up |
| hsa-miR-151a-3p | hsa-mir-151a | 4.769749212 | 2.990544659 | Up |

| miRNA name | precursor | Fold Change T1/N1 | Fold Change T2/N2 | Direction of change |
|---|---|---|---|---|
| hsa-miR-183-5p | **hsa-mir-183** | 20.03425235 | 17.04563573 | Up |
| hsa-miR-205-5p | **hsa-mir-205** | 14.92682996 | 5.746883622 | Up |
| hsa-miR-21-3p | **hsa-mir-21** | 11.17629621 | 2.581143517 | Up |
| hsa-miR-98-5p | **hsa-mir-98** | 3.019229747 | 5.633611035 | Up |
| hsa-let-7b-5p | **hsa-let-7b** | 2.102740307 | 4.321044032 | Up |
| hsa-miR-141-3p | **hsa-mir-141** | 10.55120146 | 2.056804463 | Up |
| hsa-miR-182-5p | **hsa-mir-182** | 24.16001436 | 12.97609881 | Up |
| hsa-miR-92a-3p | **hsa-mir-92a-2** | 5.629857838 | 2.286613948 | Up |
|  | **hsa-mir-92a-1** | 5.964733862 | 2.266651834 |  |
| hsa-miR-191-5p | **hsa-mir-191** | 3.327191237 | 5.037676123 | Up |
| hsa-let-7a-5p | **hsa-let-7a-3** | 2.074203797 | 3.111619867 | Up |
|  | **hsa-let-7a-2** | 2.075618201 | 3.108941695 | Up |
|  | **hsa-let-7a-1** | 2.076021712 | 3.108625093 |  |

**Table 3  Down expression of miRNA in renal pelvis cancer compared to adjacent normal tissue using next-generation sequencing.**

| miRNA name | precursor | Fold change T1/N1 | Fold change T2/N2 | Direction of change |
|---|---|---|---|---|
| hsa-miR-99a-5p | **hsa-mir-99a** | −4.218629767 | −40.88795732 | Down |
| hsa-miR-145-3p | **hsa-mir-145** | −4.030566925 | −4.695092025 | Down |
| hsa-miR-125b-5p | **hsa-mir-125b-1** | −3.344198175 | −16.626662 | Down |
| hsa-miR-125b-5p | **hsa-mir-125b-2** | −3.352884586 | −16.58782609 | Down |
| hsa-miR-145-5p | **hsa-mir-145** | −4.520230974 | −5.234484965 | Down |
| hsa-miR-100-5p | **hsa-mir-100** | −2.517396655 | −30.5010168 | Down |
| hsa-miR-451a | **hsa-mir-451a** | −3.800722042 | −20.86300667 | Down |
| hsa-let-7c-5p | **hsa-let-7c** | −2.028677204 | −11.93390878 | Down |
| hsa-miR-143-3p | **hsa-mir-143** | −2.352348002 | −3.043986991 | Down |

demonstrated that the PDE5 inhibitor sildenafil which is clinically used for treating erectile dysfunction diseases, promotes melanoma cell invasion and growth and increases melanoma risk (*Arozarena et al., 2011*; *Li et al., 2014*). Another study revealed that the cGMP pathway may have crosstalk with the MAPK pathway to affect the pathophysiology and therapy of melanoma. PDE5 inhibitors enhance melanoma cell growth by activating cGMP pathways both in vitro and in vivo, thus indicating a link between PDE5 and cGMP pathways in melanoma cells (*Dhayade et al., 2016*). The cGMP pathway also affects various reactions in tumor microenvironments, such as blood supply, angiogenesis, inflammation, and immune reactions (*Fajardo, Piazza & Tinsley, 2014*). Another theory proposed that PDE5A is related to hypoxia in tumor microenvironments. When tumor cells are under hypoxic conditions, microphthalmia-associated transcription factors are inhibited, reducing the expression of PDE5A, and promoting the proliferation and metastatic phenotype of melanoma (*Houslay, 2016*). Based on previous research and assumptions about correlations between PDE5A and melanoma, PDE5A—selected from our NGS and bioinformatics analyses—may be considered a crucial factor influencing UTUC tumor cell progression. In our in vitro study, PDE5A demonstrated lower mRNA

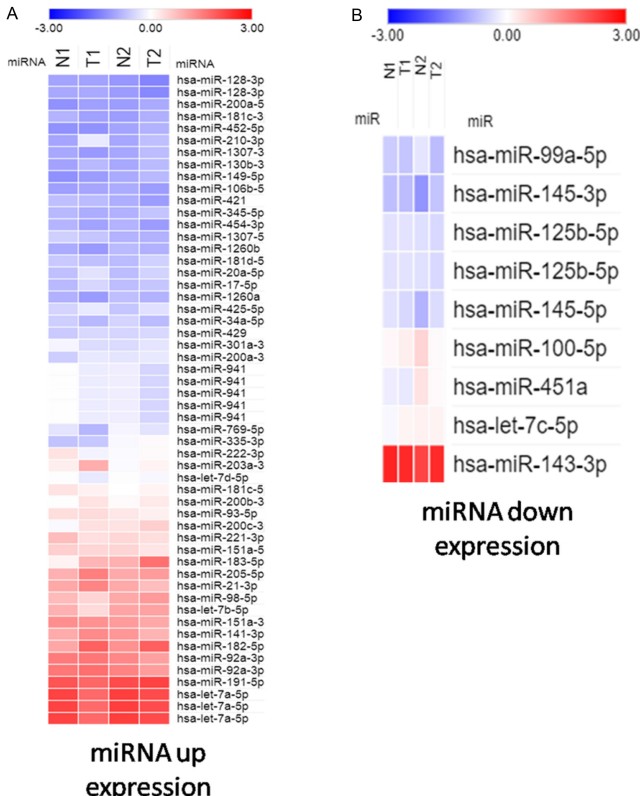

**Figure 1** Heatmap exhibiting significant difference expression of miRNAs. (A) miRNA up expression. (B) miRNA down expression.

and protein expression in advanced UC cell lines. Results from the comparison of normal tissue and tumor specimens from 20 patients also revealed lower expression in tumor specimens. Therefore, we hypothesized that PDE5A plays the role of a tumor suppressor in UC.

Reversion-inducing cysteine-rich protein with kazal motifs (RECK), which is a tumor suppressor, regulates matrix metalloproteinases (MMPs) by breaking down the extracellular matrix (ECM). The integrity of the ECM, which is maintained by RECK, is associated with tumor invasion and metastasis. When RECK inhibits MMPs, the basement membrane remains intact and inhibits tumor angiogenesis to prevent blood vessels from supplying to tumor cells. Studies have mentioned the role of RECK in tumors including lung, breast, prostate, oral, digestive tract, liver, and pancreatic cancers (*Nagini, 2012*). Because evidence suggests that the RECK gene is related to oncogenesis, it may be a therapeutic target. If a therapeutic means of enhanceing RECK expression is developed, it may be valuable in improving prognosis and impeding tumor progression. Another theory regarding RECK relates to hypoxic microenvironments, which interact with hypoxia-inducible factor (HIF)-1$\alpha$ prompting RECK downregulation with histone deacetylase (HDAC)1 to silence the RECK gene (*Zhu et al., 2017*).

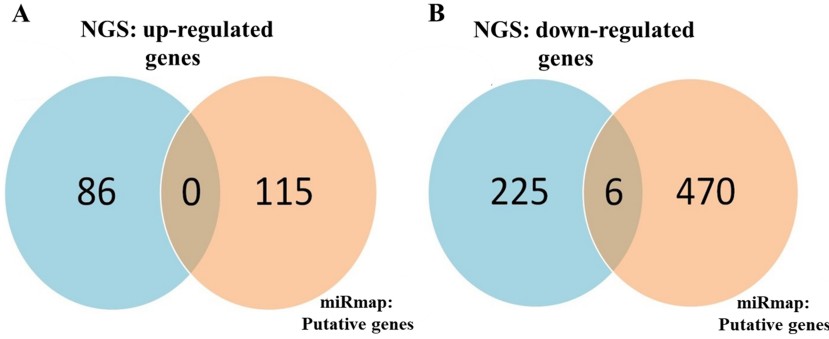

**Figure 2 Identification of potential candidate genes through a combination of next-generation sequencing (NGS) and miRmap databases.** (A) NGS analysis identified 86 upregulated genes and 115 putative genes using miRmap analysis. No potential candidates for upregulated genes were identified after interaction. (B) NGS analysis identified 225 downregulated genes and 470 putative genes through miRmap analysis. Six significant differentially downregulated candidate genes were recognized.

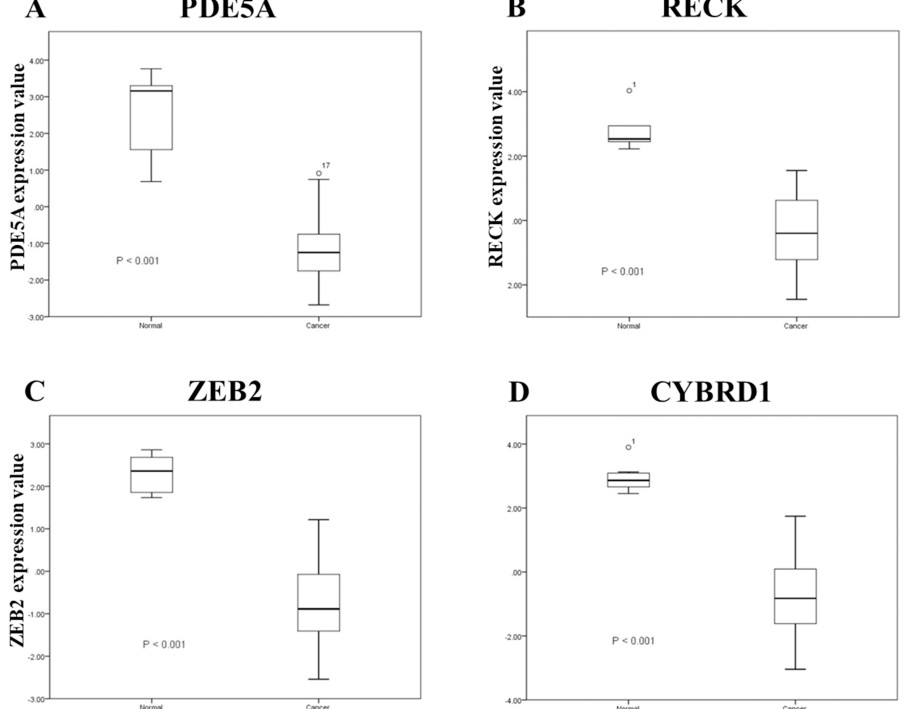

**Figure 3 A related array (Gene Expression Atlas accession: GSE19915) was used to analyze significant expression.** Expression of A: PDE5A, B: RECK, C: ZEB2, and D: CYBRD1 between urothelial carcinoma specimens and urothelial tissues.

**Table 4  Candidate genes with putative upstream miRNA and differentially expressed genes from the NGS database.**

| Up-regulated microRNA | Target down-regulated mRNA | Gene name | Fold-change (T1 vs. N1) | Fold-change (T2 vs. N2) |
|---|---|---|---|---|
| hsa-miR-181c-5p | PDE5A | phosphodiesterase 5A | 3.92 | 5.23 |
| hsa-miR-33b-3p | PDE5A | | 10.57 | 5.56 |
| hsa-miR-98-3p | PDE5A | | 3.39 | 2.41 |
| hsa-miR-200b-3p | RECK | reversion-inducing-cysteine-rich protein with kazal motifs | 9.66 | 5.98 |
| hsa-miR-200c-3p | RECK | | 12.08 | 6.29 |
| hsa-miR-34a-5p | RECK | | 3.42 | 2.17 |
| hsa-miR-429 | RECK | | 7.73 | 4.34 |
| hsa-miR-141-3p | ZEB2 | zinc finger E-box binding homeobox 2 | 10.55 | 2.06 |
| hsa-miR-200a-3p | ZEB2 | | 10.05 | 4.17 |
| hsa-miR-200b-3p | ZEB2 | | 9.66 | 5.98 |
| hsa-miR-200c-3p | ZEB2 | | 12.08 | 6.29 |
| hsa-miR-429 | ZEB2 | | 7.73 | 4.34 |
| hsa-miR-106a-5p | CYBRD1 | cytochrome b reductase 1 | 5.28 | 2.65 |
| hsa-miR-106b-5p | CYBRD1 | | 6.60 | 2.81 |
| hsa-miR-17-5p | CYBRD1 | | 10.48 | 4.79 |
| hsa-miR-20a-5p | CYBRD1 | | 11.62 | 6.66 |
| hsa-miR-652-5p | CYBRD1 | | 3.14 | 2.78 |
| hsa-miR-93-5p | CYBRD1 | | 5.66 | 3.23 |
| hsa-miR-452-5p | CYBRD1 | | 5.80 | 5.31 |

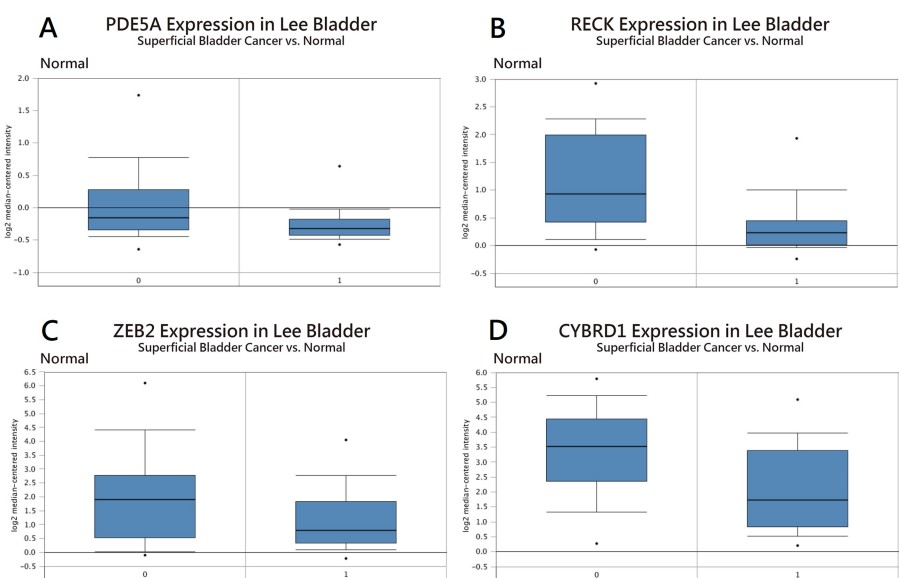

**Figure 4  Expression in bladder cancer compared with normal tissue using the Oncomine database.**
Analysis of (A) PDE5A, (B) RECK, (C) ZEB2, and (D) CYBRD1 expression.

**Network 1: Observation 1: UTUC mRNA down: Observation 1**

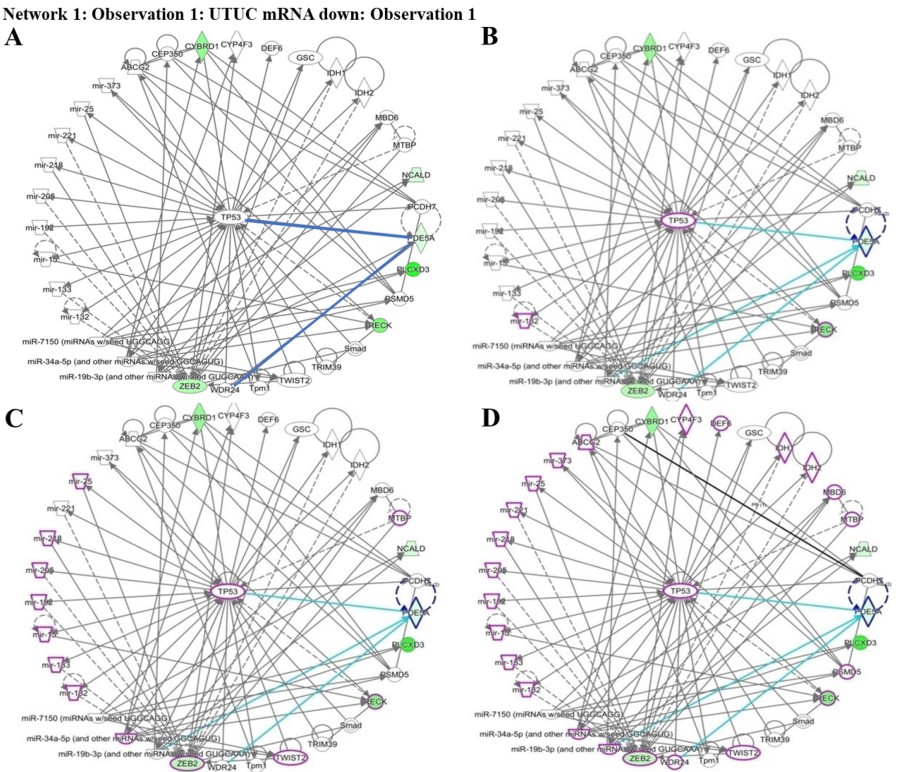

**Figure 5** **Network prediction for PDE5A, RECK, ZEB2, and CYBRD1 involved in urothelial carcinoma analyzed using the Ingenuity Pathway Analysis database.** (A) The four candidate genes were associated with TP53. (B) From the cancer invasion network, RECK—indicated by a purple circle—is a potential gene associated with cancer invasion. (C) From the cancer metastasis network, RECK and ZEB2—indicated by purple circles—play critical roles in cancer metastasis. (D) From the urogenital cancer network, RECK and ZEB2—indicated by purple circles—were demonstrated to be involved in urogenital cancer pathways.

In addition, RECK promotes angiogenesis through the mediation of the vascular endothelial growth factor, which stimulates angiogenesis and is essential for development and differentiation (*Alexius-Lindgren et al., 2014*). RECK affects not only tumor progression but also tissue architecture remodelling in embryonic development. Although the influence of RECK has been studied in various cancers, its role in UTUC remains unclear. Further investigation is required to identify potential prognostic or therapeutic targets of RECK in UTUC.

Cytochrome b reductase 1 (CYBRD1) is a member of the cytochrome b(561)family and is distributed mainly within duodenum cells. The function of CYBRD is the reduction of $Fe^{3+}$ to $Fe^{2+}$, regulation of ferrous metabolism, absorption, and transportation. A study on breast cancer detected duodenal cytochrome b (DCYTB) on the surface of breast cancer epithelial cells. The expression of DCYTB is associated with breast cancer survival, high-grade tumors demonstrated significantly lower expression of DCYTB, and at high expression levels of DCYTB, patients had better prognoses. The expression of this gene also reduces tumor cell adhesion and metastasis by diminishing the activation of focal adhesion

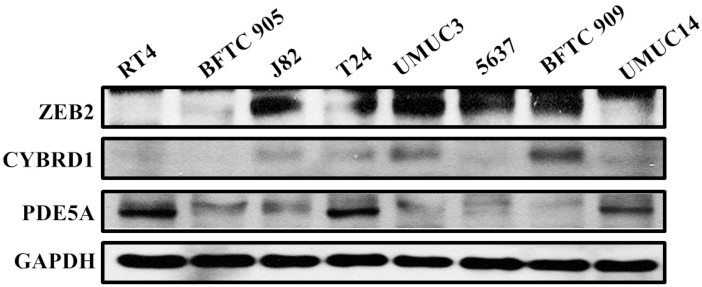

**Figure 6 Protein expression analyzed using western blot analysis.** Antibodies against human ZEB2, CY-BRD1, PDE5A, and GAPDH were employed as primary antibodies.

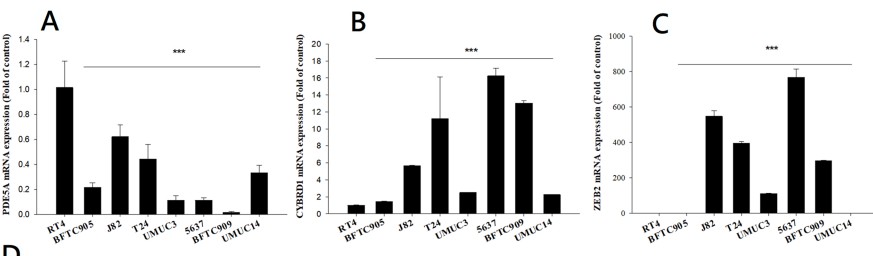

D

| qPCR | PDE5A (Fold of control) | CYBRD1 (Fold of control) | ZEB2 (Fold of control) |
|---|---|---|---|
| RT4 | $1.00 \pm 0.21$ | $1.00 \pm 0.05$ | $1.00 \pm 0.20$ |
| BFTC905 | $0.22 \pm 0.04^{***}$ | $1.46 \pm 0.04^{***}$ | $0.12 \pm 0.07^{***}$ |
| J82 | $0.62 \pm 0.09^{***}$ | $5.65 \pm 0.07^{***}$ | $548.55 \pm 31.12^{***}$ |
| T24 | $0.44 \pm 0.12^{***}$ | $11.21 \pm 4.90^{***}$ | $397.39 \pm 8.65^{***}$ |
| UMUC3 | $0.11 \pm 0.03^{***}$ | $2.50 \pm 0.03^{***}$ | $111.40 \pm 2.66^{***}$ |
| 5637 | $0.11 \pm 0.02^{***}$ | $16.24 \pm 0.92^{***}$ | $768.38 \pm 46.19^{***}$ |
| BFTC909 | $0.02 \pm 0.01^{***}$ | $13.02 \pm 0.28^{***}$ | $297.04 \pm 2.94^{***}$ |
| UMUC14 | $0.69 \pm 0.11^{***}$ | $2.26 \pm 0.01^{***}$ | $0.33 \pm 0.06^{***}$ |

**Figure 7 mRNA expression in UC and UTUC cells.** mRNA expression of (A) PDE5A, (B) CYBRD1, and (C) ZEB2 in UC and UTUC cells. J82, T24, 5637, and BFTC 909 cells exhibited higher expression of CY-BRD1 and ZEB2 but lower expression of PDE5A than RT4 cells. (D) Expression of fold of control levels.

kinase (*David et al., 2017*). Western blotting and qPCR analysis of cultured cells revealed that activation of the DCYTB gene is also regulated by Hif-2α (*Luo et al., 2014*). Further research is necessary to clarify whether the gene is a novel biomarker for UTUC.

Further searching for predictive upstream regulatory miRNAs from miRmap, TargetScan, target sites of miR-181c-5p on PDE5A were positioned 3767-3773 and 3986-3992 of PDE5A 3′ UTR. The role of miR-181c in cancer has also been validated in pancreatic cancer. Chen et al. demonstrated that high expression of miR-181c was associated with poor prognosis and survival. Furthermore, it induced cancer cell chemoresistance by inactivating the Hippo signaling pathway (*Chen et al., 2015*). A recent pancreatic cancer–bearing mouse model confirmed the hypothesis that it reverses the effect of multidrug resistance by

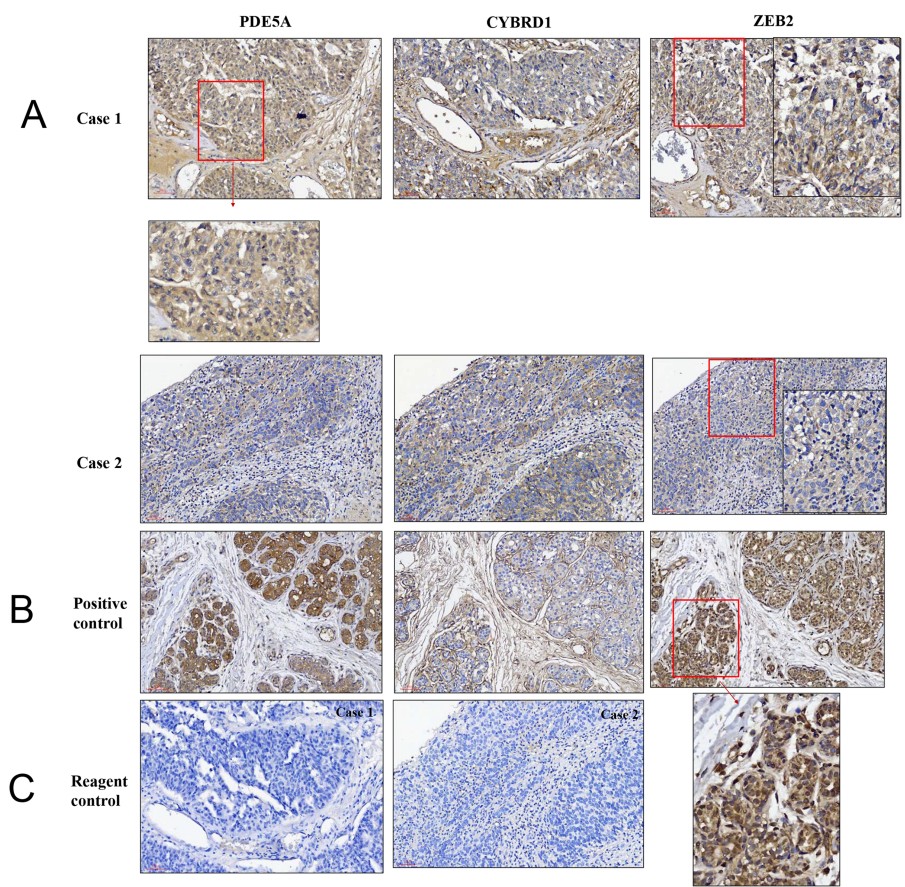

**Figure 8** **Illustration of histopathological analysis of sections from surgical resected human UTUC tissue.** (A) It shows different levels of localization for PDE5A, CYBRD1, and ZEB2 staining in 200× magnification. Scale bar, 100 μm. (B) Positive control = breast tumor tissue with higher PDE5A expression. (C) Reagent control = without primary antibody.

downregulating miR-181c-5p (*Gao et al., 2018*). Another candidate miRNA, miR-200c-3p which is at positions 126-132 and 1207-1214 of RECK 3′ UTR, has been investigated in several cancer types and is associated with cancer progression and metastasis by regulating epithelial–mesenchymal transition. However, whether the effect of regulatory mechanisms can translate to clinical survival in patients with UTUC still requires elucidation (*Wang et al., 2013*).

Due to the advent of NGS technologies over the past few years, identification of specific target genes has become a feasible strategy for stratifying the risk of disease and offering a targeted oncological treatment. A recent review article discussed the molecular characterization of UTUC using NGS analysis and demonstrated that different gene expression levels or mutational profiles present different risks of cancer, which can be target for prognostic or therapeutic strategies. If gene expression is the next era of precision therapy, combining expression profiling with potential candidate genes, including our NGS dataset, may increase the accuracy to predict cancer survival or progression. Furthermore,

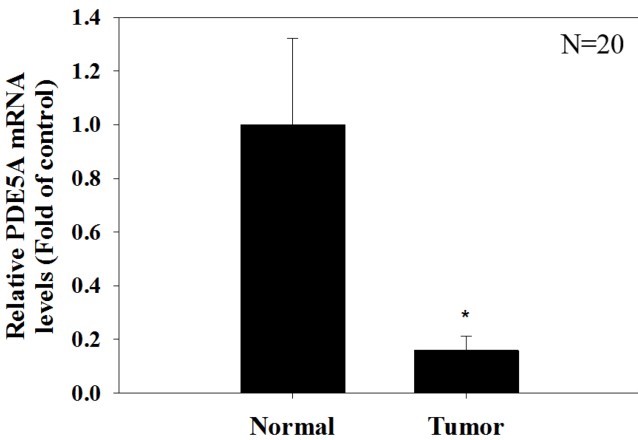

**Figure 9** **Paired UTUC tissues (normal and tumor) from 20 patients were evaluated for PDE 5A expression through real-time PCR.** GAPDH was used as an internal control. The relative expression of PDE5 in tumors was significantly lower than that in normal UTUC tissues.

**Table 5** **The clinical characteristics of 20 patients.**

|  | Number |
|---|---|
| Gender | Male: 11 |
|  | Female: 9 |
| Age (mean, year) | 65.72 |
| Tumor side | Left: 13 |
|  | Right: 7 |
| Tumor location | Ureter: 3 |
|  | Renal pelvis: 17 |
| Pathology Grade | Low grade: 2 |
|  | High grade: 18 |
| Pathology T stage | T1: 6 |
|  | T2: 4 |
|  | T3: 9 |
|  | T4: 1 |

it will also be interesting to conduct further research on different patient characteristics between countries and ethnicities (*Hassler et al., 2020*).

In conclusion, our study revealed that PDE5A, RECK, ZEB2, and CYBRD1 are predictive candidate genes in UTUC through NGS and bioinformatics analyses. The expression of these genes was lower in cancer specimens than in normal tissues. Research indicates that PDE5A, RECK, and CYBRD1are tumor suppressor genes involved in angiogenesis, hypoxia microenvironment, and metabolism. These candidate genes are associated with cancer progression and survival. In vitro studies further clarified the expression of PDE5A, CYBRD1, and ZEB2. PDE5A is also a potential tumor suppressor gene, as identified by comparing the expression levels between normal and tumor specimens from 20 patients.

Future studies with large sample size and combined candidate gene identification will be crucial to translate these findings to precision medicine strategies.

### Funding

This study was supported by grants from the Ministry of Science and Technology (MOST 106-2314-B-037-029), the Kaohsiung Medical University Hospital (KMUH 106-6R53), Kaohsiung Municipal Ta-Tung Hospital (kmtth-108-R003) and partially by the Kaohsiung Medical University Grant (KMU-KI109002), the Regenerative Medicine and Cell Therapy Research Center Grant (KMU-TC108A02), and the Cohort Research Center (KMU-TC108B07). The funders had no role in study design, data collection and analysis, decision to publish, or preparation of the manuscript.

### Grant Disclosures

The following grant information was disclosed by the authors:
The Ministry of Science and Technology: MOST 106-2314-B-037-029.
Kaohsiung Medical University Hospital: KMUH 106-6R53.
Kaohsiung Municipal Ta-Tung Hospital: kmtth-108-R003.
Kaohsiung Medical University Grant: KMU-KI109002.
Regenerative Medicine and Cell Therapy Research Center Grant: KMU-TC108A02.
Cohort Research Center: KMU-TC108B07.

### Competing Interests

The authors declare there are no competing interests.

### Author Contributions

- Hsiang-Ying Lee and Bi Wen Yeh conceived and designed the experiments, performed the experiments, analyzed the data, prepared figures and/or tables, authored or reviewed drafts of the paper, and approved the final draft.
- Ching-Chia Li, Ya-Ling Hsu, Hung-Lung Ke, Chun-Nung Huang, Chien-Feng Li and Wen-Jeng Wu conceived and designed the experiments, authored or reviewed drafts of the paper, and approved the final draft.
- Wei-Ming Li and Hsin-Chih Yeh analyzed the data, authored or reviewed drafts of the paper, and approved the final draft.
- Po-Lin Kuo conceived and designed the experiments, performed the experiments, authored or reviewed drafts of the paper, and approved the final draft.

### Human Ethics

The following information was supplied relating to ethical approvals (i.e., approving body and any reference numbers):

The Kaohsiung Medical University Hospital granted Ethical approval to carry out the study within its facilities (KMUHIRB-E(I)-20170018; KMUH107-7R56).

## Data Availability

Data is available at NCBI GEO: GSE159824.

## Supplemental Information

Supplemental information for this article can be found online at http://dx.doi.org/10.7717/peerj.11343#supplemental-information.

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
