# Peer review of "Identification of potential genes in upper tract urothelial carcinoma using next-generation sequencing with bioinformatics and in vitro analyses"

_PeerJ, doi:10.7717/peerj.11343_

## Round 0.1 · original submission · Major Revisions

The two expert reviewers have each made a number of important recommendations. I hope you see that their comments are intended to be helpful criticisms and I ask that you revise your manuscript accordingly, answering each and every issue in the revised manuscript.

Reviewer 1 ·

Basic reporting

Literature references are sufficient and updated.
Number of tables should be reduced.

Experimental design

The authors address an interesting issue on one of the less studied urological cancers, UTUC. Literature need more reports on this issue. However, the manuscript must be improve before its publication.

Validity of the findings

Conclusions are not supported by the results.

Additional comments

In this manuscript the authors aim to investigate potential biomarkers and molecular mechanisms to determine suitable diagnostic and therapeutic strategies. The authors address an interesting issue on UTUC. However there are a few questions to be answered.


Abstract: The abstract does not reflex the study design, it should be rewritten. It has to be clear what is performed and with what samples. In results section the authors mention bladder cancer, but what samples? Sample size does not appear. It is mentioned diagnostic but in the conclusions prognostic is referred.

Methodology: Initially two tumour samples are analyzed (NGS). Why a T1 and T3 tumours are selected? These are tumours with a very different progression risk. Subsequently, several searches with different platforms are performed. Which has been the criteria used?

Results: L204 Table 2. Please, group miRNA by families, there are several from the same family.
Fig1. Are the miRNA differentially expressed between tumour and healthy samples? Please, clarify.
L236 This figure is not understood
L247 Fig 6 is irrelevant. Please skip it.
L252 Need to talk about RECK gene?
Why is decided to analyze selected genes in cell lines and not in patients samples?It is not clear to me if there are 10 or 20 samples.

Discussion: Why the authors refer to progression? It is difficult to extrapolate the results of a in vitro study with a reduced number of samples.
Conclusion: It should not refer to progression because of the study design.

Reviewer 2 ·

Basic reporting

1. The language needs to be polished. There are some repetitions. The method should be included in the result part, at least the way of expression should be changed, For example, In result "we analysed two pairs of patient specimens to compare renal pelvis tumours and adjacent nontumour urothelium tissues through NGS". It is unnecessary.
2. There are some misspellings, such as in Fig.9 levels of localisation, it should be localization. Furthermore, the text seems unrelated to the localization.

Experimental design

1. In the Immunohistochemical staining part: incubated with an antibody against PDE5A,
190 CYBRD1, and ZEB2 at 37°Cfor 1 hour. The detailed information of the antibodies should be provided as clone No, dilution, and company.
2.For the Next-generation sequencing, what are the sequencing depth and coverage?
3.Several urothelial cancer cell lines (RT4, BFTC905, J82, T24, UMUC3, BFTC 909 ) were used in this study. It seems that they are all bladder urothelial cancer cell lines. UTUC cell line would be best for the validation.

Validity of the findings

1.In the result part, NGS analysis identified 86 upregulated genes and 115 putative genes by using miRmap analysis. It showed that no potential candidates for upregulated genes were identified after interaction. The Fig. 2A should be revised since the interaction number is zero. It is kind of strange that there is no potential upregulated gene. Please provide more evidence.
2.Fig 9 of histopathological analysis of different levels of localization for PDE5A, CYBRD1, and ZEB2 staining in human UTUC samples. The quality of pictures are bad, please provide better pictures in higher magnification. The signal of Zeb2 should be in the nuclei of tumor cells, while it is hard to discern the location of signals.
3.The discussion is kind of lacking focus. Since the main result was derived from the mRNA NGS, it should be compared with the other NGS study in urothelial carcinoma.

---

## Round 0.2 · Minor Revisions

The reviewer comments will require only revision of the text. I agree with the reviewer that the statements that they mention are too strongly worded and therefore inaccurate, or are not directly supported by your study.

I do not expect the revision will be difficult to achieve if you follow the reviewer's recommendations.

Reviewer 2 ·

Basic reporting

no comment

Experimental design

no comment

Validity of the findings

There are some statements not accurate.
1. In the abstract, "lower PDE5A expression in tumor specimens was also demonstrated in paired UTUC tissue" should be by using paired UTUC and normal tissues, lower PDE5A expression was also demonstrated in tumor specimens.
2. "Our findings suggest these candidate genes may play crucial roles in UTUC progression". The results only suggest these genes may play some roles in UTUC tumorgenesis. Furthermore the statement "may be potential targets both as prognostic factors and for therapeutic strategies" is lacking powerful evidence. There is no survival information.
3. In the discussion part, "combining expression profiling with potential candidate genes, including our NGS dataset, may increase the accuracy rate when diagnosing UTUC". What is the point of using these genes in diagnosing UTUC? how? by blood or urine test? If they are available, this kind of description should be avoided.

Additional comments

no comment

---

## Round 0.3 · accepted · Accept

Thank you for submitting your revision.